

# Oxygen-limited metabolism in the methanotroph *Methylomicrobium buryatense* 5GB1C

Alexey Gilman[1,*], Yanfen Fu[1,*], Melissa Hendershott[2], Frances Chu[3], Aaron W. Puri[1], Amanda Lee Smith[4], Mitchell Pesesky[1], Rose Lieberman[5], David A.C. Beck[1,6] and Mary E. Lidstrom[1,7]

[1] Department of Chemical Engineering, University of Washington, Seattle, WA, United States of America
[2] Allen Institute for Cell Science, Seattle, WA, USA
[3] InBios, Seattle, WA, USA
[4] Zymo Genetics, Seattle, WA, USA
[5] Department of Biology, George Washington University, Washington, D.C., USA
[6] eScience Institute, University of Washington, Seattle, WA, USA
[7] Department of Microbiology, University of Washington, Seattle, WA, USA
[*] These authors contributed equally to this work.

Corresponding author
Mary E. Lidstrom, lidstrom@uw.edu

## ABSTRACT

The bacteria that grow on methane aerobically (methanotrophs) support populations of non-methanotrophs in the natural environment by excreting methane-derived carbon. One group of excreted compounds are short-chain organic acids, generated in highest abundance when cultures are grown under $O_2$-starvation. We examined this $O_2$-starvation condition in the methanotroph *Methylomicrobium buryatense* 5GB1. The *M. buryatense* 5GB1 genome contains homologs for all enzymes necessary for a fermentative metabolism, and we hypothesize that a metabolic switch to fermentation can be induced by low-$O_2$ conditions. Under prolonged $O_2$-starvation in a closed vial, this methanotroph increases the amount of acetate excreted about 10-fold, but the formate, lactate, and succinate excreted do not respond to this culture condition. In bioreactor cultures, the amount of each excreted product is similar across a range of growth rates and limiting substrates, including $O_2$-limitation. A set of mutants were generated in genes predicted to be involved in generating or regulating excretion of these compounds and tested for growth defects, and changes in excretion products. The phenotypes and associated metabolic flux modeling suggested that in *M. buryatense* 5GB1, formate and acetate are excreted in response to redox imbalance. Our results indicate that even under $O_2$-starvation conditions, *M. buryatense* 5GB1 maintains a metabolic state representing a combination of fermentation and respiration metabolism.

## IMPORTANCE

The ability of methanotrophs to excrete short-chain acids has implications for environmental consumption of the potent greenhouse gas methane. Under the $O_2$-starvation conditions similar to those in many natural environments where methanotrophs

are found, formate and acetate are major products, with lactate and succinate also generated. This methane-derived carbon is involved in supporting a community of non-methanotrophs in such natural environments. This work also suggests approaches for maximizing excretion of specific products for bioconversion applications of methanotrophs.

## INTRODUCTION

Methanotrophs are a group of bacteria able to grow on methane as sole carbon and energy source (*Whittenbury, Phillips & Wilkinson, 1970*). They play an important role in natural habitats, retaining methane carbon in ecosystems and mitigating emissions of this potent greenhouse gas (*Knief, 2015*; *Chistoserdova, 2015*). The subgroup of methanotrophs requiring $O_2$ for activation of methane (the aerobic methanotrophs) are important in mitigating methane emissions from soil and aquatic environments (*Ruff et al., 2015*). Aerobic methanotrophs often live at the low end of the $O_2$ gradient in their natural environments (*Knief, 2015*), creating selective pressure to develop mechanisms for coping with $O_2$-starvation. In addition, such bacteria are also of interest for developing commercial gas-to-liquid processes, converting methane to fuels and chemicals (*Kalyuzhnaya, Puri & Lidstrom, 2015*).

Recently, it has been shown that a gamma-proteobacterial methanotroph, *Methylomicrobium alcaliphilum* 20Z, contains a highly efficient version of the ribulose monophosphate cycle for formaldehyde assimilation, that could theoretically allow for a fermentation type of metabolism, with $O_2$ used for activating the methane molecule, but not as a terminal electron acceptor (*Kalyuzhnaya et al., 2013*). Evidence was provided that genes encoding enzymes of such a pathway are widespread in gamma-proteobacterial methanotrophs, and are transcriptionally up-regulated when cells are cultured under $O_2$-starvation conditions, concomitant with excretion of putative fermentation end products (*Kalyuzhnaya et al., 2013*). These results are intriguing, both for the potential of methanotrophs to cross-feed non-methanotrophs with methane-derived carbon in natural communities (*Radajewski, McDonald & Murrell, 2003*; *Oshkin et al., 2015*), and for the potential to manipulate methanotrophic metabolism to generate excreted products (*Kalyuzhnaya, Puri & Lidstrom, 2015*).

The genome of *M. buryatense* 5GB1 predicts a set of genes that could be involved in fermentation of formaldehyde, via the ribulose monophosphate cycle and the EMP pathway and/or XFP pathway (Fig. 1; *De la Torre et al., 2015*; *Henard, Smith & Guarnieri, 2017*), similar to such genes identified in *M. alcaliphilum* 20Z (*Kalyuzhnaya et al., 2013*). These genes predict a set of end products that could be generated, including formate, acetate, lactate, succinate, and $H_2$.

We have examined this mode of $O_2$-starved metabolism in a different gamma-proteobacterial methanotroph, *Methylomicrobium buryatense* 5GB1, for which formate, acetate, and lactate excretion have been reported (*Gilman et al., 2015*; *Henard et al., 2016*; *Henard, Smith & Guarnieri, 2017*). This methanotroph has become an attractive model system for carrying out basic research studies of methanotrophs, with a genome

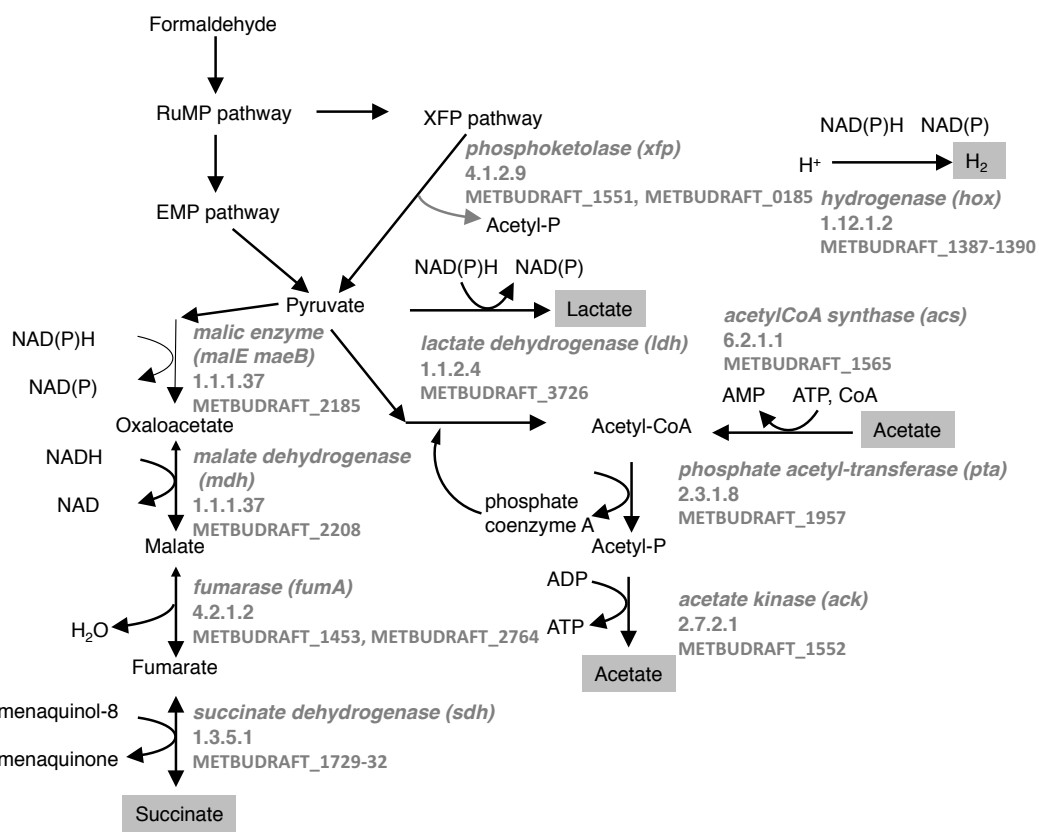

**Figure 1** **Predicted pathways for generation of excreted products in *M. buryatense* 5GB1.** Enzymes and gene designations are listed.

sequence (*Khmelenina et al., 2013*), a set of genetic tools (*Puri et al., 2015*; *Yan et al., 2016*; *Henard et al., 2016*), a genome-scale metabolic model (*De la Torre et al., 2015*), a metabolic flux database (*Fu, Li & Lidstrom, 2017*) and a set of transcriptomic, metabolomic, and bioreactor datasets available (*De la Torre et al., 2015*; *Gilman et al., 2015*; *Fu, Li & Lidstrom, 2017*; *Henard, Smith & Guarnieri, 2017*). We hypothesized that, like *M. alcaliphilum* 20Z, *M. buryatense* 5GB1 would switch to a primarily fermentative metabolic state under $O_2$-starvation, increasing its excretion of organic end products. We have used a variety of approaches to determine how the excretion products are generated under $O_2$-starvation, and have used metabolic modeling coupled to measured parameters to predict the structure of the metabolic network occurring during this mode of growth.

## MATERIALS AND METHODS

### Strains and culture conditions

*M. buryatense* 5GB1C is a derivative of *M. buryatense* 5GB1 created by deliberate curing of its 80 kbp plasmid. *M. buryatense* 5GB1C was not found to have a growth defect or significant changes in chromosomal gene expression in the absence of its plasmid, but this did allow for genetic manipulation (*Puri et al., 2015*). For this study we used *M. buryatense*

5GB1 for most wild-type experiments, while all gene knockouts were completed in the 5GB1C background. Two biological replicates of wild-type *M. buryatense* 5GB1C were grown in the bioreactor under slow growth $O_2$-starvation conditions for comparison to the aa3 cytochrome oxidase mutant strain. *M. buryatense* 5GB1 and its derivatives were grown in a modified NMS medium with methane at 30 °C, as previously described (*Puri et al., 2015*). *Methylobacter tundripaludum* 31/32 and *Methylomonas* sp. LW13 (*Kalyuzhnaya et al., 2015*) were grown in NMS medium at 18 °C and 30 °C, respectively, as previously described for *Methylobacter tundripaludum* 21/22 (*Puri et al., 2017*). The *M. buryatense* 5G genome sequence is deposited in GenBank/EMBL under the accession numbers AOTL01000000, KB455575, and KB455576 (*Khmelenina et al., 2013*). *M. buryatense* 5GB1 cultures grown in vials for $O_2$-starvation experiments were inoculated at 0.05 $OD_{600}$ from an overnight culture and the vial was given a normal headspace of 25% $CH_4$, 75% air. Cultures were incubated with shaking at 30 °C for five days. For growth curve experiments, cultures were set up the same way, and samples were taken at various time points to measure $OD_{600}$ and excretion products. For *M. tundripaludum* 31/32, incubation temperature was 18 °C.

## Product measurements

Measurements of excreted organic acids were carried out either by NMR, as previously described (*Kalyuzhnaya et al., 2013*), or by ion chromatography, as follows. 1 ml of cell culture was taken from vial, and centrifuged at 13,000 rpm for 5 min. The supernatant was carefully transferred into a new syringe and filtered through a 0.2 μm syringe filter (Millipore syringe filter; Thermo Fisher Scientific, Waltham, MA, USA). 600–650 μl filtered supernatant was transferred into PolyVials (Thermo Fisher Scientific, Waltham, MA, USA) for injection. Analysis was carried out on an ICS-1600 Ion chromatography system equipped with a ICE-AS6 9 × 250 mm column. 225 μl was injected into the system, eluted with 1 ml/min of 0.5 mM heptafluorobutyric acid (HFBA; Thermo Fisher Scientific, Waltham, MA, USA). The column was regenerated with 5 mM tetrabutylammonium hydroxide (TBAOH, Thermo Fisher Scientific, Waltham, MA, USA) aqueous solution. All cultures were assessed in two separate biological replicates and results are reported as an average.

## Bioreactor

Bioreactor experiments were carried out in a New Brunswick BioFlo 310 bioreactor (Eppendorf, Inc. Enfield, CT, USA), and off gas samples were measured using Shimadzu gas chromatograph GC2014 (Shimadzu Scientific Instruments, Inc. Columbia, MD, USA) as described previously (*Gilman et al., 2015*), using premixed gas tanks (20% $CH_4$, 5% $O_2$, 75% $N_2$) and a dilution rate (equal to the growth rate at steady state) of 0.03 $hr^{-1}$.

## Steady state $^{13}$C tracer experiments

Steady state $^{13}$C tracer experiments were performed on the aa3 cytochrome oxidase mutant strain in methane, as described previously (*Fu, Li & Lidstrom, 2017*). 5 ml of preculture was inoculated from plate, cultured in 25% $^{13}$C methane (Sigma) to full growth, and transferred into 50 ml fresh NMS2 medium with OD600 = 0.01 with 25% $^{13}$C methane (Sigma) until mid-log phase. The cell culture was quenched, extracted and measured using LC-MS/MS with previously described method (*Fu, Li & Lidstrom, 2017*).

## RNAseq

RNA was extracted as previously described (*Chu & Lidstrom, 2016*). RNA sequencing was performed by GENEWIZ (South Plainfield, NJ, USA) using Illumina HiSeq2500 $1 \times 50$ (single ended) reads. The raw reads from the sequencing facility were aligned to the annotated *M. buryatense* 5G genome as downloaded from JGI's IMG on July 14, 2017 (*Markowitz et al., 2014*). Alignment was performed using BWA version 0.7.12-r1044 using the BWA-MEM algorithm and default parameters (*Li & Durbin, 2010*). The alignments were post-processed into sorted BAM files with SAMTools version 1.2-232-g87cdc4a (*Li et al., 2009*). Reads were attributed to open reading frames (ORFs) using the *htseq-count* tool from the 'HTSeq' framework version 0.8.0 in the 'intersection-nonempty' mode (*Anders, Pyl & Huber, 2015*). Differential abundance analysis was performed with DESeq2 1.2.10 (*Anders & Huber, 2010*; *Anders et al., 2013*) using R 3.3.1.

Genes were considered to be differentially expressed if there was an average change of greater than 2-fold when comparing normalized counts as well as an adjusted *p*-value of less than $1E - 05$ (*Anders & Huber, 2010*).

## Mutants and genetic manipulations

Gene knockout constructs were generated using assembled PCR products that were electroporated into *M. buryatense* 5GB1C, as previously described (*Chu & Lidstrom, 2016*; *Yan et al., 2016*). Double mutants were constructed with the same approach into unmarked single mutant *M. buryatense* 5GB1C strain. Plasmids were constructed by Gibson assembly (*Gibson et al., 2009*).

## Flux balance analysis with Cobrapy

The genome scale model published earlier (*De la Torre et al., 2015*) was used in this study with a few modifications. The constraint of flux partition between the EMP and ED pathways was removed by replacing reaction PYK and EDA with reactions that produced regular pyruvate. The secretion reactions of organic acids (specifically, formate, acetate, succinate and lactate) were decoupled from the biomass equation 'BIOMASS_M5GB1'. The respiration summary reaction was bound to 0, and was replaced with full electron transport chain reactions. The methane oxidation reaction (pMMO) was modified with ubiquinol as electron acceptor, methanol oxidation reaction (MXA) and nitrate reductase were also modified accordingly. CobraPy (*Ebrahim et al., 2013*) (version 0.4.1) was used in this study for flux balance analysis with cglpk as solver. Python scripts and input model files (in .xml files) are available in the Zenodo repository: https://zenodo.org/badge/latestdoi/93199234.

## RESULTS

### Excretion products in vial-grown cultures under $O_2$-starvation

*M. buryatense* 5GB1 was grown in closed vials under $O_2$-starvation conditions, formate, acetate, and succinate were detected, lactate, hydroxybutyrate, and $H_2$ were not detectable (Fig. 2). Compared to previously reported *M. buryatense* 5GB1 bioreactor cultures, the acetate was over 10-fold higher in the vial cultures, but the formate was similar to the values in the bioreactor cultures (Fig. 2; *Gilman et al., 2015*). Since the *Methylomicrobium*

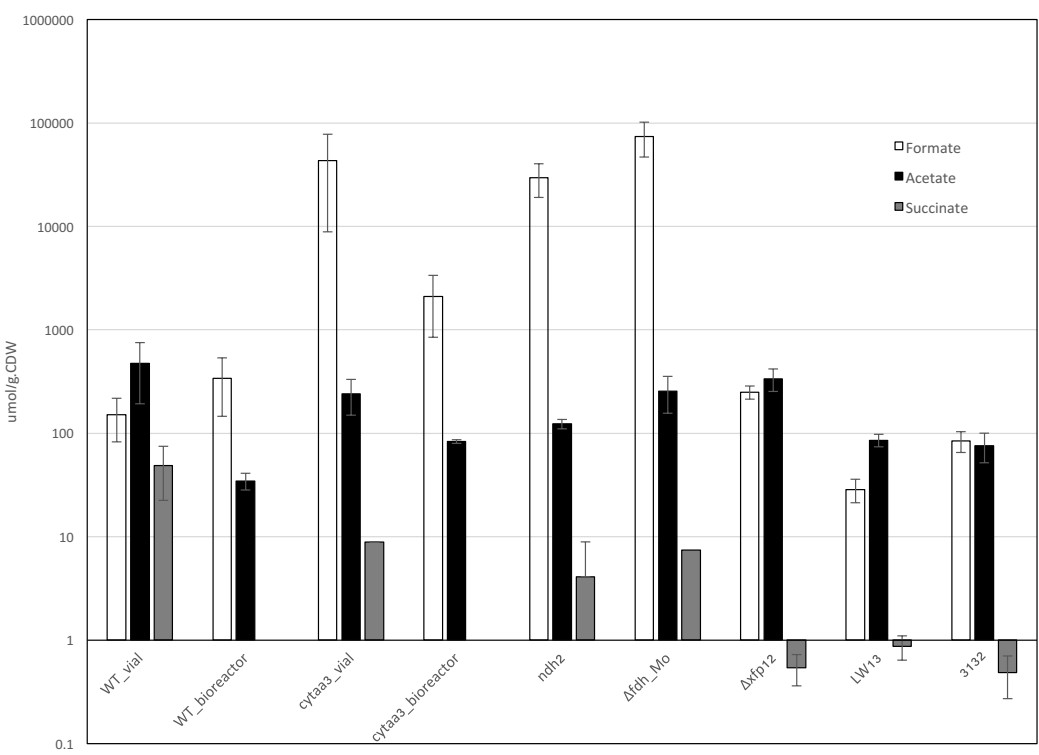

**Figure 2** Excreted products profile for *M. buryatense* 5GB1 (WT) and the aa3 cytochrome oxidase mutant (Δcytaa3) in $O_2$-limited vial and bioreactor cultures, for *M. buryatense* 5GB1C mutants in NADH dehydrogenase 2 (Δndh2), molybdenum-containing formate dehydrogenase (Δfdh_Mo), and a double mutant in the phosphoketolase homologs (Δxfp12) in $O_2$-limited vial cultures, and for *Methylomonas sp.* LW13 (LW13) and *Methylobacter tundripaludum* 31/32 (3132) in $O_2$-limited vial cultures. Lactate and hydroxybutyrate were not detected. $H_2$ was tested in the wild-type *M. buryatense*, and was not detected.

strains tested for these excretion products so far are both haloalkaliphiles (*Kaluzhnaya et al., 2001*), we also assessed two different gamma-proteobacterial methanotrophs isolated from a freshwater lake, *Methylobacter tundripaludum* 31/32 and *Methylomonas* strain LW13 (*Kalyuzhnaya et al., 2015*), and showed that these strains generate formate, acetate, and succinate under these growth conditions, but lactate and hydroxybutyrate were not detected (Fig. 2).

## Excretion products and $O_2/CH_4$ uptake ratio in bioreactor cultures

As reported previously, formate, acetate, and lactate had been detected in bioreactor cultures of *M. buryatense* 5GB1, with similar values in cultures grown on methane at maximum growth rate (0.22–0.24 $hr^{-1}$) and under methane- and $O_2$-limitation grown at about half the maximum growth rate (*Gilman et al., 2015*). *M. buryatense* 5GB1 was tested in this study for excreted products under the slower growth conditions used for *M. alcaliphilum* 20Z, in an $O_2$-limited bioreactor operated at a growth rate of 0.035–0.036 $hr^{-1}$. Samples were taken at steady-state for excreted products and RNAseq analysis, and the $O_2/CH_4$ ratio was determined. The excreted products and $O_2/CH_4$ ratios were compared to the values

**Table 1** Oxygen-limited metabolism in the methanotroph *Methylomicrobium* buryatense 5GB1C Bioreactor data summary.

| Experiment ID | FM86 | FM87 | FM88 | FM90 |
|---|---|---|---|---|
| Strain | 5GB1 WT | 5GB1 WT | 5GB1 ΔCyt aa3 | 5GB1 ΔCyt aa3 |
| Dilution rate ($h^{-1}$) or growth rate | 0.036 | 0.035 | 0.034 | 0.035 |
| Cell density (gCDW $L^{-1}$) | 0.32 | 0.28 | 0.25 | 0.21 |
| Dissolved oxygen (mg $L^{-1}$) | 0.22 | 0.25 | 0.25 | 0.26 |
| Oxygen uptake (mmol $h^{-1}$) | 0.92 | 0.94 | 0.95 | 0.94 |
| Methane uptake (mmol $h^{-1}$) | 0.86 | 0.88 | 0.88 | 0.89 |
| Specific oxygen uptake (mmol gCDW$^{-1}$ $h^{-1}$) | 2.9 | 3.1 | 3.9 | 4.5 |
| Specific methane uptake (mmol gCDW$^{-1}$ $h^{-1}$) | 2.7 | 2.9 | 3.6 | 4.2 |
| $O_2/CH_4$ uptake ratio | 1.1 | 1.1 | 1.1 | 1.1 |
| Inlet gas composition | 20% $CH_4$, 5%$O_2$, 75%$N_2$ | 20% $CH_4$, 5%$O_2$, 75%$N_2$ | 20% $CH_4$, 5%$O_2$, 75%$N_2$ | 20% $CH_4$, 5%$O_2$, 75%$N_2$ |
| Formate (µmol gCDW$^{-1}$) | 479.1 | 203.1 | 1216.3 | 2995.2 |
| Acetate (µmol gCDW$^{-1}$) | 30.2 | 39.2 | 85.7 | 81.0 |

previously published for *M. buryatense* 5GB1 $O_2$-limited and methane-limited cultures at a higher growth rate (0.12 $hr^{-1}$) (Table 1; *Gilman et al., 2015*). The bioreactor cultures did not show an increase in the excreted products compared to full $O_2$ bioreactor cultures, and the $O_2/CH_4$ ratios were similar to those previously published for the faster growth $O_2$-limited cultures (*Gilman et al., 2015*).

## Transcriptomics

RNA samples from the slower growth $O_2$-limited cultures were used for RNAseq analysis and compared to RNA samples from the previously published bioreactor conditions noted above ($O_2$-limited and $CH_4$-limited, both at a growth rate of 0.12 $hr^{-1}$; Table 2; Table S1). Only a few of the genes predicted to be involved in a fermentation type metabolism (Fig. 1) showed significant changes in transcription, when compared to either the $O_2$-limited or $CH_4$-limited cultures (Table 3). The gene most strongly regulated was that for bacteriohemerythrin (MBUTDRAFT_0310), encoding a protein proposed to be involved in $O_2$-scavenging (*Chen et al., 2012*) and shown to be induced in *M. alcaliphilum* 20Z under fermentation conditions (*Kalyuzhnaya et al., 2013*). This gene was upregulated over two orders of magnitude when compared to the high $O_2$ condition ($CH_4$-limited), 2-fold when the faster growth $O_2$-limitation was compared to the $CH_4$-limitation condition, and 10-fold, when the slower-growth $O_2$-limitation condition was compared to the faster growth $O_2$-limitation condition. This result suggests that the slower growth $O_2$-limitation condition was more $O_2$-stressed than the higher growth $O_2$-limitation condition. Other genes in this list that showed a pattern of upregulation in response to low $O_2$ were those encoding the three subunits of a predicted ba3-type cytochrome oxidase and one of the subunits of the aa3 cytochrome oxidase. One of the few gene clusters predicted to be

Gilman et al. (2017), *PeerJ*, DOI 10.7717/peerj.3945

**Table 2** RNAseq analysis of genes involved in genes predicted to have a potential role in fermentation metabolism (see Table 3).

| Conditions | | | Slow growth O2-limited vs. fast growth O2-limited | | Slow growth O2-limited vs. fast growth CH4-limited | | Fast growth O2-limited vs. fast growth CH4-limited | | Cytochrome oxidase aa3 vs. WT (both slow growth O2-limited) | |
|---|---|---|---|---|---|---|---|---|---|---|
| Gene | Enzyme | Locus tag | Fold change | Adjusted *p* values | Fold change | Adjusted *p* values | Fold change | Adjusted *p* values | Fold change | Adjusted *p* values |
| **xfp1** | **phospho-ketolase** | **METBUDRAFT_1551** | 2.1 | **0.031** | 1.6 | 0.258 | **0.7** | **0.013** | 1.7 | 0.219 |
| xfp2 | phospho-ketolase | METBUDRAFT_0185 | 1.3 | 0.404 | 1.3 | 0.478 | 1 | 0.904 | 1.1 | 0.997 |
| ack | acetate kinase | METBUDRAFT_1552 | 0.9 | 0.674 | 0.8 | 0.65 | 1 | 0.839 | 1.2 | 0.948 |
| acs | acetylCoA synthase | METBUDRAFT_1565 | 1.6 | 0.141 | 1.3 | 0.516 | 0.8 | 0.227 | 1.2 | 0.902 |
| pta | phosphotrans-acetylase | METBUDRAFT_1957 | 1.1 | 0.819 | 1.1 | 0.802 | 1 | 0.837 | 1 | 1 |
| ldh | lactate dehydrogenase | METBUDRAFT_3726 | 1 | 0.95 | 0.8 | 0.665 | 0.9 | 0.33 | 1 | 1 |
| mdh | malate dehydrogenase | METBUDRAFT_2208 | 1 | 0.934 | 0.9 | 0.862 | 1 | 0.825 | 1.1 | 0.997 |
| Malic enzyme | malic enzyme | METBUDRAFT_2185 | 1 | 0.916 | 1 | 0.918 | 0.9 | 0.584 | 1.1 | 0.997 |
| **hox** | **hydrogenase** | **METBUDRAFT_1387** | **2.6** | **1.0 E−04** | 1.5 | 0.181 | **0.6** | **2.5 E−05** | 1.5 | 0.422 |
| | | **METBUDRAFT_1388** | **2** | **0.01** | 1.4 | 0.364 | **0.7** | **0.004** | 1.5 | 0.51 |
| | | **METBUDRAFT_1389** | **1.9** | **0.021** | 1.5 | 0.258 | 0.8 | 0.07 | 1.3 | 0.858 |
| | | METBUDRAFT_1390 | 1.5 | 0.206 | 1.5 | 0.255 | 1 | 0.866 | 1.2 | 0.901 |
| ndh1 | NADH dehydrogenase | METBUDRAFT_1319 | 1.2 | 0.534 | 1.3 | 0.556 | 1 | 0.871 | 1.1 | 0.997 |
| **ndh2a** | **NADH dehydrogenase** | **METBUDRAFT_2827** | **0.5** | **0.03** | 0.6 | 0.091 | 1.2 | 0.45 | 1.2 | 0.907 |
| ndh2b | | METBUDRAFT_2828 | 0.6 | 0.175 | 0.9 | 0.785 | 1.4 | 0.139 | 1.2 | 0.898 |
| petA | bc1 complex | METBUDRAFT_2502 | 1.4 | 0.33 | 1.2 | 0.755 | 0.8 | 0.385 | 1.3 | 0.794 |
| petB | | METBUDRAFT_2503 | 1.7 | 0.147 | 1.4 | 0.401 | 0.8 | 0.358 | 1.4 | 0.678 |
| petC | | METBUDRAFT_2504 | 1.6 | 0.138 | 1.3 | 0.555 | 0.8 | 0.128 | 1.2 | 0.919 |
| **ctaC** | **cytochrome oxidase (aa3-type)** | **METBUDRAFT_0311** | **1.9** | **0.044** | **2.1** | **0.011** | 1.1 | 0.501 | (deleted) | |
| ctaD | | METBUDRAFT_0312 | 1.5 | 0.317 | 1.8 | 0.1 | 1.2 | 0.207 | (deleted) | |
| ctaG | | METBUDRAFT_0313 | 1.4 | 0.129 | 1.4 | 0.201 | 1 | 0.978 | (very low expression) | 3.57 E−09 |
| ctaE | | METBUDRAFT_0314 | 1.1 | 0.786 | 1.1 | 0.92 | 0.9 | 0.791 | (very low expression) | 1.57 E−07 |
| **cbaA** | **cytochrome oxidase (ba3-type)** | **METBUDRAFT_1311** | 1.4 | 0.24 | **2.1** | **0.005** | **1.5** | **0.001** | 1.3 | 0.856 |

**Table 2** (*continued*)

| | Conditions | | Slow growth O2-limited vs. fast growth O2-limited | | Slow growth O2-limited vs. fast growth CH4-limited | | Fast growth O2-limited vs. fast growth CH4-limited | | Cytochrome oxidase aa3 vs. WT (both slow growth O2-limited) | |
|---|---|---|---|---|---|---|---|---|---|---|
| Gene | Enzyme | Locus tag | Fold change | Adjusted *p* values | Fold change | Adjusted *p* values | Fold change | Adjusted *p* values | Fold change | Adjusted *p* values |
| ***cbaB*** | | **METBUDRAFT_1312** | 1.4 | 0.302 | **2.1** | **0.011** | **1.5** | **0.002** | 1.3 | 0.858 |
| ***cbaD*** | | **METBUDRAFT_1313** | 1.3 | 0.358 | **2.1** | **0.005** | **1.5** | **0.001** | 1.1 | 0.997 |
| ***gnd1*** | **6-phosphogluconate dehydrogenase (NADP)** | **METBUDRAFT_3313** | **1.6** | **0.04** | 1.5 | 0.118 | 0.9 | 0.693 | 1.5 | 0.431 |
| *gnd2* | 6-phosphogluconate dehydrogenase (NAD) | METBUDRAFT_3982 | 0.9 | 0.674 | 1.1 | 0.847 | 1.3 | 0.089 | 0.9 | 0.997 |
| *mtd1* | methylene H4MPT de-hydrogenase | METBUDRAFT_1893 | 1.3 | 0.306 | 1.3 | 0.496 | 1 | 0.811 | 1.2 | 0.997 |
| *mtd2* | methylene H4MPT de-hydrogenase | METBUDRAFT_1894 | 1.1 | 0.821 | 1.1 | 0.855 | 1 | 0.95 | 1.1 | 0.997 |
| *fdh1a* | formate dehydrogenase (tungsten) | METBUDRAFT_0831 | 0.7 | 0.22 | 0.6 | 0.103 | 0.9 | 0.67 | 1.2 | 0.997 |
| *fdh1b* | | METBUDRAFT_0832 | 0.6 | 0.224 | 0.6 | 0.163 | 1 | 0.977 | 1.3 | 0.728 |
| *fdh2a* | formate dehydrogenase (molybdenum) | METBUDRAFT_2829 | 0.9 | 0.912 | 0.9 | 0.868 | 1 | 0.874 | 1.1 | 0.997 |
| *fdh2b* | | METBUDRAFT_2830 | 0.8 | 0.315 | 0.7 | 0.17 | 0.9 | 0.485 | 1.1 | 0.997 |
| ***fdh2c*** | | **METBUDRAFT_2831** | 1.5 | 0.115 | 0.8 | 0.4 | **0.5** | **1.64 E−06** | 1.2 | 0.989 |
| ***bhr*** | **Bacterio-hemerythrin** | **METBUDRAFT_0310** | **9.9** | **1.39 E−11** | **391.4** | **1.96 E−198** | **2.2** | **0.001** | 1.2 | 0.883 |
involved in fermentation that showed consistent and statistically significant up-regulation under the slower growth condition was that for the first three genes encoding hydrogenase (MBUTDRAFT1387-89), with the genes showing 1.9–2.6 fold change compared to the faster growth $O_2$-limited condition. However, since $H_2$ was not detected under these conditions, this expression change did not result in net $H_2$ evolution. These genes showed a significant but small decrease in expression when the faster growth $O_2$-limited culture was compared to the faster growth methane-limited culture. Other genes showing an upregulation response to the slower growth condition compared to the faster growth $O_2$-limited condition were those for *xfp1*, encoding one of the phosphoketolase genes (METBUDRAFT_1551) at 2.1 fold, and one of the genes encoding a 6-phosphogluconate dehydrogenase homolog (METBUDRAFT_3313), 1.6 fold. One of the subunits of one of the NADH dehydrogenase clusters (METBUDRAFT_2827) showed downregulation under slow growth, suggesting it might be growth-rate regulated.

The entire set of genes was assessed to determine whether any recognizable metabolic trends could be extracted from the gene expression pattern (Table S1). The slower growth $O_2$-limited cultures showed significant induction of the entire set of nitrogenase genes (METBUDRAFT_4082-4154) while in the other RNAseq datasets the expression levels were near background, suggesting nitrogen-stress at the slower growth condition. Further work showed that only when the nitrate was increased to 3-fold the normal amount was nitrogenase repressed in the bioreactor cultures at the slower growth condition, but that level of nitrate caused growth inhibition (data not shown). Therefore, it was not used for further studies. However, no other general metabolic trends could be discerned from this dataset.

## Mutant phenotypes

In order to assess the role of specific gene products in the generation of excreted organic acids, a set of 25 deletion mutations were generated (Table 3). These included genes predicted to be involved in generating $H_2$, formate, acetate, lactate, and succinate, genes predicted to be involved in utilization of NADH, and genes predicted to be involved in generation of NADH. The latter two categories were included to assess the role of NADH balance in excretion phenotypes. In addition, attempts were made to generate a double mutant defective in acetate kinase and acetylCoA synthase, to assess the role of this interconversion in acetate production, but we were unable to obtain a null mutant.

Only one mutant showed a noticeable growth defect on agar plates, the mutant deleted in the aa3 cytochrome oxidase (*ctaCD*). In liquid culture with sufficient $O_2$, this mutant had a maximum growth rate of 0.06 $hr^{-1}$ and a maximum OD600 of 0.2–0.4. Excretion products were tested for vial cultures of all mutants incubated for five days. Only three mutants showed a significant change in excretion products compared to the wild type (Fig. 2), the aa3 cytochrome oxidase mutant, the mutant in the predicted molybdenum-dependent formate dehydrogenase (*fdh2*) and the mutant in one of the predicted NADH dehydrogenase genes (*ndh2*). In the last two cases, formate excretion increased over two orders of magnitude, acetate decreased about 2-fold, and succinate excretion decreased about an order of magnitude, while in the aa3 cytochrome oxidase mutant, formate excretion increased

**Table 3** *M. buryatense* 5GB1C mutant characteristics: bold-face, mutants with phenotypes; *italics, no null mutants obtained.*

| Gene(s) deleted | Enzyme | Locus tag(s) | Phenotypes | Category |
|---|---|---|---|---|
| *ack* | acetate kinase | METBUDRAFT_1552 | none | fermentation product: acetate |
| *acs* | acetylCoA synthase | METBUDRAFT_1565 | none | fermentation product: acetate |
| *ack/acs* | *acetate kinase/acetylCoA synthase (double)* | *METBUDRAFT_1552/ MET-BUDRAFT_1565* | *no null mutants* | *fermentation product: acetate* |
| *pta* | phosphotransacetylase | METBUDRAFT_1957 | none | fermentation product: acetate |
| *xfp1* | phosphoketolase | METBUDRAFT_1551 | none | fermentation product: acetate |
| *xfp2* | phosphoketolase | METBUDRAFT_0185 | none | fermentation product: acetate |
| *xfp1/xfp2* | phosphoketolase (double) | METBUDRAFT_1551/ MET-BUDRAFT_0185 | none | fermentation product: acetate |
| *ldh* | lactate dehydrogenase | METBUDRAFT_3726 | none | fermentation product: lactate |
| *mdh* | malate dehydrogenase | METBUDRAFT_2208 | none | fermentation product: succinate |
| *malic_enzyme* | malic enzyme | METBUDRAFT_2185 | none | fermentation product: succinate |
| *hox* | hydrogenase | METBUDRAFT_1387-1390 | none | fermentation product: $H_2$ |
| *ndh1* | NADH dehydrogenase | METBUDRAFT_1319 | none | NADH utilization |
| **ndh2** | **NADH dehydrogenase** | **METBUDRAFT_2827** | **formate up; acetate and succinate down** | NADH utilization |
| **ndh1/ndh2** | **NADH dehydrogenase (double)** | **METBUDRAFT_1319/ MET-BUDRAFT_2827** | **formate up, acetate and succinate down** | NADH utilization |
| *petABC* | bc1 complex | METBUDRAFT_2502-2504 | none | NADH utilization |
| **ctaCD** | **cytochrome oxidase aa3-type** | METBUDRAFT_0311-0312 | **formate up; acetate and succinate down; growth rate down** | NADH utilization |
| *cba* | cytochrome oxidase cba-type | METBUDRAFT_1311-1313 | none | NADH utilization |
| *gnd1* | 6-phosphogluconate dehydrogenase (NADP) | METBUDRAFT_3313 | none | NADH generation |
| *gnd2* | 6-phosphogluconate dehydrogenase (NAD) | METBUDRAFT_3982 | none | NADH generation |
| *gnd1/gnd2* | 6PGDH (double) | METBUDRAFT_3313/ METBUDRAFT_3982 | none | NADH generation |
| *mtd1* | *methylene H4MPT dehydrogenase* | METBUDRAFT_1893 | *no null mutants* | *NADH generation* |
| *mtd2* | *methylene H4MPT dehydrogenase* | METBUDRAFT_1894 | *no null mutants* | *NADH generation* |
| *fdh1ab* | formate dehydrogenase (tungsten) | METBUDRAFT_0831-0832 | none | NADH generation |
| **fdh2** | **formate dehydrogenase (molybdenum)** | **METBUDRAFT_2831** | **formate up; acetate and succinate down** | NADH generation |
| *fdh1/fdh2* | **formate dehydrogenase double** | **METBUDRAFT_0831-0832/METBUDRAFT_2831** | **formate up; acetate and succinate down; growth rate down** | NADH generation |
| *bhr* | bacteriohemerythrin | METBUDRAFT_0310 | none | $O_2$ starvation |

about an order of magnitude, and acetate and succinate excretion decreased 3- to 5-fold (Fig. 2).

Since three of the functions tested had two predicted homologs in the genome, double mutants were generated for 6-phosphogluconate dehydrogenase (*gnd*), NADH dehydrogenase (*ndh*), formate dehydrogenase (*fdh*) and phosphoketolase (*xfp*). Only the
*fdh1/fdh2* double mutant showed a growth defect on agar plates. In liquid culture, it grew with a growth rate of $0.04 \pm 0.01$ hr$^{-1}$. In the case of the first two double mutants, the excretion phenotypes in vial cultures were no different from the single mutants. In the case of the *xfp1/xfp2* double mutant, the formate and acetate levels did not change significantly, but the succinate levels dropped almost two orders of magnitude. The excretion profile of the *fdh1/fdh2* double mutant was similar to that of the *fdh2* single mutant.

## aa3 cytochrome oxidase mutant strain analysis

Given the severe growth defect of the mutant deleted in the aa3 cytochrome oxidase, it might be expected that this strain could only grow via a fermentation-like metabolism, without significant aerobic respiration. Further experiments were carried out to assess this possibility. The aa3 cytochrome oxidase mutant strain was grown in the bioreactor under $O_2$-limited conditions similar to the wild type, with a growth rate of 0.034–0.035 hr$^{-1}$. At this growth rate it was possible to achieve steady-state culture of this mutant strain, and at steady-state conditions the levels of formate and acetate were both higher than the wild type (Fig. 2). The $O_2$/$CH_4$ ratio was 1.1, similar to that for the wild-type (Table 1). RNA samples from the steady-state culture were used for RNAseq analysis but few differences above 2-fold with significant p-values were observed comparing the mutant to the wild type (Table S1), other than those involved in nitrogen metabolism (Table S1). None of the genes predicted to be involved in a fermentation-type metabolism showed significant changes in the aa3 cytochrome oxidase mutant compared to the wild type (Table 3).

## FBA modeling

An existing genome-scale FBA model (*De la Torre et al., 2015*) was updated to include the fermentation reactions shown in Fig. 1 as well as expanded to include individual electron transport chain reactions (Table S2). Secreted products were uncoupled from the biomass equation, to facilitate changes to these values. Bioreactor values for methane uptake rates were used to simulate specific cases using COBRApy (Table 4). In the first case, no other constraints were applied, and the model predicts flux ratios similar to those for unrestricted methane growth, decreased to accommodate the lower methane uptake rates, with a predicted growth rate average of $0.042 \pm 0.002$ hr$^{-1}$. When the additional constraint of the $O_2$/$CH_4$ ratio was applied to accommodate the restricted $O_2$ availability, the model predicts a growth rate average of $0.035 \pm 0.001$ hr$^{-1}$, similar to the experimental growth rate average of 0.0355 hr$^{-1}$. Finally, the constraints of the measured formate, acetate, and succinate values were applied, and the results suggested a growth rate average of 0.0335 $\pm$ 0.001 hr$^{-1}$, also similar to the experimental values. The *aa3* oxidase mutant strain was also modeled using the bioreactor values from Table 2 and the full set of constraints. The results predicted a growth rate average of $0.046 \pm 0.004$ hr$^{-1}$, significantly greater than the experimental growth rate average of 0.0345 hr$^{-1}$.

Since this mutant strain appeared to carry out a fermentation-type metabolism in the presence of higher $O_2$, it presented the possibility to apply steady-state metabolic flux analysis to assess metabolic flux through the pathways downstream of the pyruvate node (*Fu, Li & Lidstrom, 2017*). Such analysis is difficult to carry out under $O_2$-limitation, due

**Table 4  Flux balance analysis results.**

| Experiment ID | FM86 | FM87 | FM88 | FM90 |
|---|---|---|---|---|
| Strain | 5GB1 WT | 5GB1 WT | 5GB1 ΔCyt aa3 | 5GB1 ΔCyt aa3 |
| Experimental growth rate ($h^{-1}$) | 0.036 | 0.035 | 0.034 | 0.035 |
| Growth rate from simulation case1 (specific methane uptake rate constraint) | 0.04 | 0.043 | NA | NA |
| Growth rate from simulation case2 (adding $O_2$ uptake rate constraint to case 1) | 0.034 | 0.036 | NA | NA |
| Growth rate from simulation case3 (adding secreted products constraints to case 2) | 0.033 | 0.034 | NA | NA |
| Growth rate from simulation case4 (ΔCyt aa3 with methane, $O_2$ uptake rate and secreted products constraint) | NA | NA | 0.043 | 0.048 |

to the requirement for gas flow-through to maintain steady-state low $O_2$ levels, and the quantities of $^{13}CH_4$ required. At higher $O_2$, it is possible to carry out this analysis in a closed vial (*Fu, Li & Lidstrom, 2017*), making metabolic flux analysis of the aa3 oxidase mutant strain feasible. Metabolic flux analysis was carried out in the *aa3* oxidase mutant strain, and it was shown that in keeping with the model predictions, the TCA cycle is complete as it is in the WT strain (*Fu, Li & Lidstrom, 2017*), contributing to 45% of *de novo* malate synthesis (Fig. 3).

## DISCUSSION

The ability of aerobic methanotrophs to excrete short-chain organic acids has implications for both environmental and industrial applications of methanotrophs (*Knief, 2015*; *Chistoserdova, 2015*; *Kalyuzhnaya, Puri & Lidstrom, 2015*). Understanding the metabolism involved in this conversion is an important step in defining how methane is consumed in natural environments, as well as moving towards cost-effective processes for methane biotechnology. In this work, we have shown that *Methylomicrobium buryatense* 5GB1 shows different excretion profiles as compared to a related gamma-proteobacteriual methanotroph *Methylomicrobium alcaliphilum* 20Z, highlighting differences in the metabolic networks of these two haloalkaliphilic methanotrophs. Although *M. alcaliphilum* 20Z increases multiple excretion products (formate, acetate, lactate, succinate, and $H_2$) in response to $O_2$-starvation (*Kalyuzhnaya et al., 2013*), *M. buryatense* 5GB1 only shows an increase for acetate. In addition, although both strains have similar genes for generating excretion products and both excrete formate, acetate, and succinate, *M. alcaliphilum* 20Z also produces $H_2$, lactate, and hydroxybutyrate, while *M. buryatense* 5GB1 does not excrete these compounds at detectable levels. Two different gamma-proteobacterial methanotrophs isolated from a freshwater lake, *Methylobacter tundripaludum* 31/32 and *Methylomonas* sp. LW13, show excretion profiles more similar to *M. buryatense* 5GB1 suggesting that *M. alcaliphilum* 20Z regulates this metabolism differently from the other

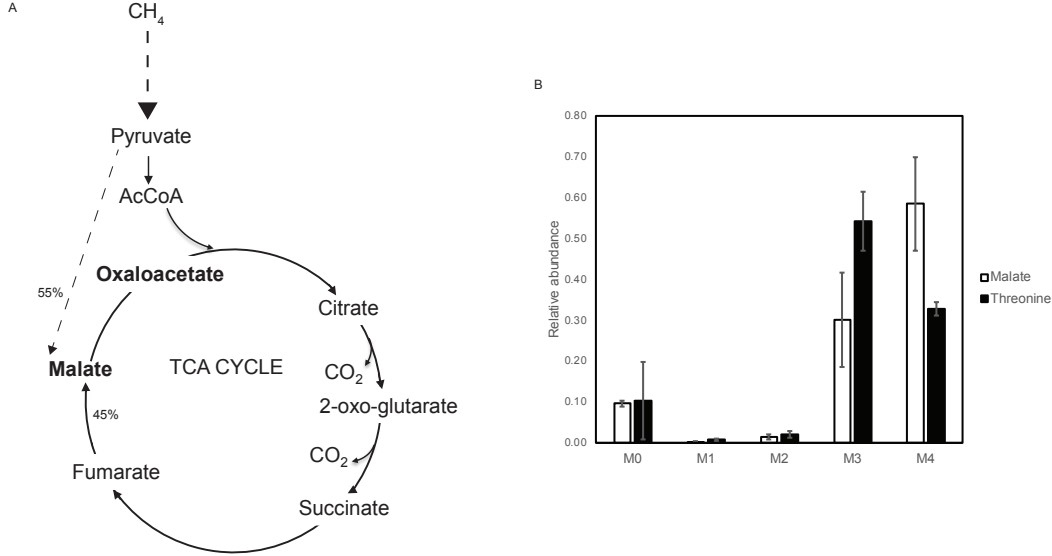

**Figure 3** **Steady state $^{13}$C tracer analysis of the aa3 cytochrome oxidase mutant strain.** (A) Metabolic subnetwork that could be deciphered using steady state $^{13}$C tracer analysis. Pathways with dashed line indicate multiple reactions combined into one reaction. Labeling pattern of metabolites in bold are measured using LC-MS/MS. TCA cycle contributes 45% of total influx into malate de novo synthesis, similar to WT. (B) Labeling patterns of malate and threonine (reflecting OAA) of the aa3 mutant strain showing different carboxylation signature.

Type I methanotrophs tested. The $O_2/CH_4$ ratio measured for *M. buryatense* 5GB1 under slower growth $O_2$-limitation (one-seventh of the maximum growth rate) is similar to that for faster growth $O_2$-limitation (one-half of the maximum growth rate), and in both cases, the results suggest a small amount of respiration. One $O_2$ is needed for each methane consumed, so any $O_2$ utilized above a 1:1 ratio is assumed to be used for respiration. The metabolic model constrained by the experimental $O_2/CH_4$ ratio predicts a significant fraction (71% ± 1%) of the ATP is generated by substrate level phosphorylation, compared to 53% ± 1% predicted with the unconstrained model. Therefore, in *M. buryatense* 5GB1 grown at low growth rate under $O_2$-limitation, the excreted organic acids are not generated in a true fermentation metabolism, but in a mixed mode of respiration and fermentation.

The aa3 cytochrome oxidase mutant strain shows a major defect in both growth rate and yield, but carries out the same relative level of respiration as the wild-type in the $O_2$-limited bioreactor. The genome contains multiple alternative predicted terminal oxidases, and it is likely that one or more of these is able to support respiration in the absence of the aa3 oxidase complex, albeit at reduced growth rate and yield. Although the model predicted a higher growth rate for the aa3 cytochrome oxidase mutant strain than that observed in the experimental work, this likely reflects either an inaccurate prediction of the bioenergetics of the alternative cytochrome oxidases used in the aa3 cytochrome oxidase mutant strain, or insufficient activity of those alternate enzymes in the cultures.

Our results show that although the genetic potential for a full fermentative metabolism exists in *M. buryatense* 5GB1, this metabolic mode does not occur under the conditions

tested. That result suggests either that the full fermentative metabolism occurs under conditions not tested, or that *M. buryatense* 5GB1 has not evolved to use a fermentation metabolism in response to low $O_2$ availability or minimal ATP production from respiration. Further work is required to determine if fermentation is commonly utilized by aerobic methanotrophs in their natural environments, and if it can be reproduced in an industrial setting.

Our results provide insights into the metabolism of formaldehyde and formate under low $O_2$ conditions (Fig. 4). Methanol dehydrogenase generates formaldehyde from methanol (*De la Torre et al., 2015*). In all of these experiments, no lanthanides were added, so the dominant methanol dehydrogenase is the calcium-dependent MxaF-type enzyme, not the lanthanide-dependent methanol dehydrogenase (Xox) (*Chu & Lidstrom, 2016*). Formaldehyde is then consumed, with two possible routes existing: (1) incorporation into the ribulose monophosphate cycle for assimilation via hexulose phosphate synthase, and/or (2) oxidation to formate by the $H_4$MPT-dependent pathway (*De la Torre et al., 2015*). It is possible to bypass formaldehyde oxidation completely, using the cyclic ribulose monophosphate pathway for $CO_2$ generation. *M. buryatense* 5GB1 contains two copies of genes encoding the key enzyme for that pathway, 6-phosphogluconate dehydrogenase. Mutation of these genes, either singly or doubly, did not generate a significant growth defect and did not change the excretion profile, suggesting that flux through this pathway is minimal under the growth conditions tested. However, it was not possible to obtain null mutants in genes of the $H_4$MPT pathway, confirming the importance of this pathway in methanotrophic metabolism in this bacterium, and confirming significant flux from formaldehyde to formate.

Formate is the major excreted organic acid, and our results suggest that formate is excreted as a result of oxidation of formaldehyde to formate at a higher flux than the oxidation of formate to $CO_2$. Regulation of formate excretion could occur at the level of export or formate dehydrogenase activity. This conclusion is supported by our findings that when the major formate consuming enzyme (formate dehydrogenase) is mutated, excreted formate increases more than 200 fold. When formate is excreted instead of being oxidized to $CO_2$, it decreases the amount of NADH formed (Fig. 4). The phenotype of the mutant in the major NADH consuming reaction, NADH dehydrogenase, was similar to that of the formate dehydrogenase mutant, suggesting the possibility that formate excretion could be linked to redox imbalance.

The severe growth defect of the aa3 oxidase mutant strain suggests that this terminal oxidase is the major one under the growth conditions tested. When the terminal electron acceptor is restricted, it would reduce the amount of NADH consumed, which might be predicted to generate a redox imbalance. As noted above, excreting a portion of formate would avoid generation of NADH from the oxidation of that formate to $CO_2$, which might partially alleviate redox imbalance.

*M. buryatense* 5GB1 contains genes for interconverting acetyl-CoA, acetyl-phosphate and acetate and also for generating acetyl-phosphate via the XFP pathway, from xylulose-phosphate. The latter pathway has been predicted to operate in *M. buyatense* 5GB1 (*Henard, Smith & Guarnieri, 2017*). Our results suggest that the excreted acetate is produced by a

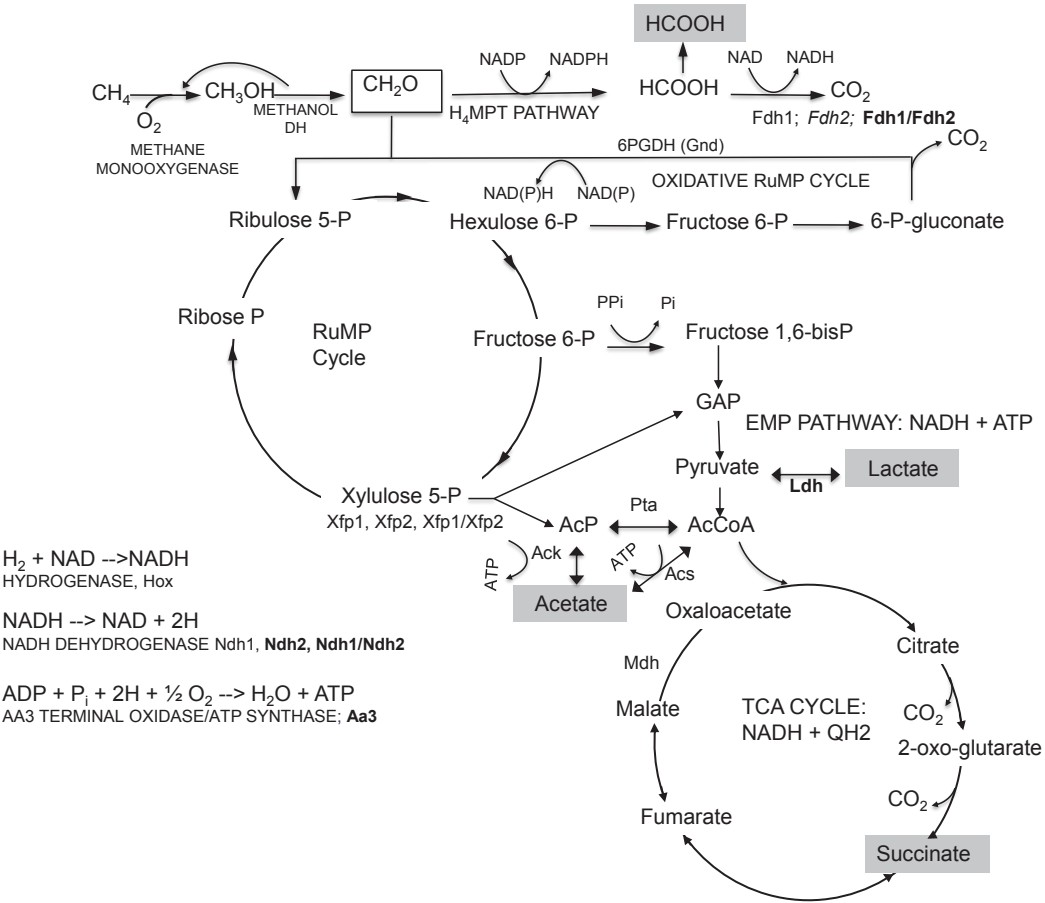

**Figure 4** Summary of central metabolism in *M. buryatense* 5GB1. GAP, glyceraldehyde- 3-P; KDPG, 2-keto-3-deoxy-6-P-gluconate; BPGA, 1,3-bisP-glycerate; PGA, 3-P-glycerate and 2-P-glycerate; PEP, phosphoenolpyruvate. Mutants with growth and excretion phenotypes in bold; those with excretion phenotypes only, in italics.

combination of the *ACK* and *ACS* pathways. The *xfp1/xfp2* double knockout has limited impact on the secretion of acetate, suggesting the acetate production route was not interrupted in this strain. The unsuccessful attempts to generate the *ack/acs* double mutant strain suggested that this interconversion between acetyl-P, acetate, and acetylCoA may be essential for growth. The Xfp route would not generate NADH, compared to the route via pyruvate dehydrogenase, and it is possible that acetate excretion is a second mechanism for managing redox balance.

The insights obtained in this work have implications for consumption of methane in natural communities. It has been shown that methane is consumed by a community of methanotrophic and non-methanotrophic bacteria, with methanotrophs excreting carbon to support the non-methanotrophic population (*Radajewski, McDonald & Murrell, 2003*; *Oshkin et al., 2015*). Recent coculture studies have demonstrated that a major interaction occurs between a methanotroph and a non-methanotrophic methylotroph isolated from

a lake sediment methane consuming community (*Krause et al., 2017*). In that study, under O$_2$-limitation conditions similar to those in which these bacteria exist in the lake sediment, the methanotroph was found to excrete methanol to support the non-methanotrophic methylotroph (*Krause et al., 2017*). Similarly, methane-starved cells of a marine methanotroph were found to excrete methanol upon recovery (*Tavormina et al., 2017*). Our results suggest that under the O$_2$-starvation conditions tested here, the methanotrophs should also excrete formate and acetate, which would be expected to support a broader population of non-methylotrophs such as those observed previously (*Oshkin et al., 2015*).

This work also suggests approaches for increasing excretion of specific compounds for biotechnology applications. Our evidence is consistent with excretion of formate and acetate as a mechanism to achieve redox balance. Correcting the imbalance is not as simple as decreasing the NADH pool, since addition of lactate dehydrogenase to *M. buryatense* 5GB1C and concomitant production of lactate at a level similar to acetate did not decrease formate + acetate excretion as might be expected (*Henard et al., 2016*). Likely, it will be necessary to assess flux through the entire metabolic network to determine network response under such conditions. The first steps to measure metabolic flux in methanotrophs have now been reported (*Fu, Li & Lidstrom, 2017*), suggesting this approach will soon be feasible as a tool for metabolic engineering. In addition, the updated metabolic model for *M. buryatense* 5GB1 reported here will provide a valuable companion tool to effectively predict modifications that will lead to desired metabolic outcomes.

## CONCLUSION

In summary, under the O$_2$-starvation conditions tested in this study, *M. buryatense* 5GB1 maintains a metabolic state representing a hybrid metabolism of fermentation and respiration. The phenotype of mutants with associated metabolic flux modeling suggested that secretion of formate and acetate could be a response to redox imbalance.

## SEQUENCE DATA

Normalized counts and computed pairwise fold changes for the RNA-seq experiments are available in Table S1. All reads files (fastq-format) and per-gene read counts were submitted to the Sequence Read Archive (SRA) and Gene Expression Omnibus (GEO), respectively, under bioproject number PRJNA396065.

## ACKNOWLEDGEMENTS

We would like to thank Alan Bohn for help in constructing the *ndh*2 and *bhr* mutants.

### Funding

This study was funded by a grant from the NSF (MCB-1409338). This work was facilitated through the use of advanced computational storage and networking infrastructure provided

by the Hyak supercomputer system supported in part by the University of Washington eScience Institute. There was no additional external funding received for this study. The funders had no role in study design, data collection and analysis, decision to publish, or preparation of the manuscript.

### Grant Disclosures

The following grant information was disclosed by the authors:
NSF: MCB-1409338.
University of Washington eScience Institute.

### Competing Interests

Melissa Hendershott is currently an employee of the Allen Institute for Cell Science, Frances Chu is currently an employee of InBios, and Amanda Lee Smith is currently an employee of Zymo Genetics.

### Author Contributions

- Alexey Gilman conceived and designed the experiments, performed the experiments, analyzed the data, wrote the paper, prepared figures and/or tables, reviewed drafts of the paper.
- Yanfen Fu conceived and designed the experiments, performed the experiments, analyzed the data, wrote the paper, prepared figures and/or tables, reviewed drafts of the paper, manuscript submission.
- Melissa Hendershott, Frances Chu and Aaron W. Puri conceived and designed the experiments, performed the experiments, analyzed the data, reviewed drafts of the paper.
- Amanda Lee Smith and Mitchell Pesesky conceived and designed the experiments, performed the experiments, analyzed the data, prepared figures and/or tables, reviewed drafts of the paper.
- Rose Lieberman performed the experiments, analyzed the data.
- David A.C. Beck conceived and designed the experiments, analyzed the data, contributed reagents/materials/analysis tools, prepared figures and/or tables, reviewed drafts of the paper.
- Mary E. Lidstrom conceived and designed the experiments, analyzed the data, wrote the paper, prepared figures and/or tables, reviewed drafts of the paper.

### Data Availability

Raw data for python scripts at Zenodo: DOI 10.5281/zenodo.842900.
Raw data for RNAseq: GEO accession number: GSE101981 (public datasets).

### Supplemental Information

Supplemental information for this article can be found online at http://dx.doi.org/10.7717/peerj.3945#supplemental-information.

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
