# Peer review of "Oxygen-limited metabolism in the methanotroph Methylomicrobium buryatense 5GB1C"

_PeerJ, doi:10.7717/peerj.3945_

## Round 0.1 · original submission · Minor Revisions

As you will see from the reviewers' reports, both are supportive of publication of your very interesting paper, however, also provide a number of suggestions for further improvement.

These mostly focus on improvements in presentation of results obtained in this study, and indicating more clearly where data are taken from previous studies.

·

Basic reporting

This is an interesting study with an appropriate background and rational for the study. Methods are clearly presented and as far as my experience allows appear to be appropriate for the study. In general I found the results to be clearly presented and the discussion and conclusions to be appropriate, however there are a number of areas within the manuscript that need improvement and are confusing.

My main concern is with the first three paragraphs of the results.
The first on Genomic analysis reads like an introduction and presents no new data and should be in the introduction.
The next two on Excretion products are confusing. They contain significant references to past work, are more discussion than results and as such it is very difficult to see what was achieved in this study. In particular references to Gilman et al (lines 209/212) suggest that work presented is from that study and not this.
I am sure there are results in these sections but it is not clear to the reader as presented, and needs to be significantly improved.

In the results the authors refer to growth rates, but in Table 1 and the methods talk about dilution rates, and then in Table 4 growth rates. There is nothing to say that they are the same thing, this needs to be clarified.

In Table 2 one of the columns is headed padj, what is this?

Experimental design

no comment

Validity of the findings

no comment

·

Basic reporting

This paper examines a type I methanotroph Methylomicrobium buryatense 5GB1 under oxygen-starvation conditions, and used a multifaceted approach to understand its oxygen-limited metabolism. The authors reported their finding in vial experiments, continuous bioreactor experiments, RNAseq, mutant testing and flux balance analysis using a previously developed model, and compared them with previously published results on the strain and a closely related strain. The paper is well-written, well-organized and easy to follow. It provided necessary background information, and conforms to PeerJ standards.

Experimental design

This paper addresses an important question, i.e., understanding type I methanotroph’s oxygen limited metabolism, as it is highly relevant not only to understand the methanotrophs’ role in ecosystems, but also to the methanotroph-based biotechnology such as converting natural gas to valuable liquid products. The hypothesis was clearly stated, experiments well designed and carried out, with clear description on different methods.

Validity of the findings

By integrating the results from different approaches, it was concluded that M. buryatense 5GB1 maintains a hybrid metabolism of fermentation and respiration, even though a genetic potential for a full fermentative metabolism exists. In addition, it was suggested that the secretion of formate and acetate could be a response to redox imbalance. The conclusion are supported by the findings reported in the paper.
Two suggestions:
1. Since the authors compared their results with previously published ones, it would be helpful to list the previously published ones in the table for easy comparison. For example, adding result from Gilman et al to Table 1 to provide a direct comparison.
2. Was produced CO2 measured? I understand it is challenging to obtain an accurate measurement of CO2 in vial experiments. But it should be easier for bioreactor experiments. If the CO2 was measured, then it would provide more information for FBA modeling, and allow a better understanding on the carbon flux shift under oxygen limited conditions.

---

## Round 0.2 · accepted · Accept

Thanks for incorporating suggested changes that further improved the manuscript. Exciting to see this interesting study being published.

·

Basic reporting

No comment

Experimental design

No comment

Validity of the findings

No comment

Additional comments

I am happy with the changes made by the authors